# Development of a Water Transmission Rate (WTR) Measurement System for Implantable Barrier Coatings

**DOI:** 10.3390/polym15112557

**Published:** 2023-06-01

**Authors:** Sébastien Buchwalder, Cléo Nicolier, Mario Hersberger, Florian Bourgeois, Andreas Hogg, Jürgen Burger

**Affiliations:** 1School of Biomedical and Precision Engineering, University of Bern, Güterstrasse 24/26, 3010 Bern, Switzerland; cleo.nicolier@unibe.ch (C.N.); mario.hersberger@unibe.ch (M.H.); juergen.burger@med.unibe.ch (J.B.); 2Graduate School for Cellular and Biomedical Sciences, University of Bern, Mittelstrasse 43, 3012 Bern, Switzerland; 3Coat-X SA, Eplatures-Grise 17, 2300 La Chaux-de-Fonds, Switzerland; bourgeois@coat-x.com (F.B.); hogg@coat-x.com (A.H.)

**Keywords:** water transmission rate (WTR), water vapor transmission rate (WVTR), gas transmission rate (GTR), parylene, barrier coatings, biomedical encapsulation

## Abstract

While water vapor transmission rate (WVTR) measurement is standardly used to assess material permeability, a system able to quantify liquid water transmission rate (WTR) measurement is highly desirable for implantable thin film barrier coatings. Indeed, since implantable devices are in contact or immersed in body fluids, liquid WTR was carried out to obtain a more realistic measurement of the barrier performance. Parylene is a well-established polymer which is often the material of choice for biomedical encapsulation applications due to its flexibility, biocompatibility, and attractive barrier properties. Four grades of parylene coatings were tested with a newly developed permeation measurement system based on a quadrupole mass spectrometer (QMS) detection method. Successful measurements of gas and water vapor and the water transmission rates of thin parylene films were performed and validated, comparing the results with a standardized method. In addition, the WTR results allowed for the extraction of an acceleration transmission rate factor from the vapor-to-liquid water measurement mode, which varies from 4 to 4.8 between WVTR and WTR. With a WTR of 72.5 µm g m^−2^ day^−1^, parylene C displayed the most effective barrier performance.

## 1. Introduction

Currently, multiple instruments are available for the measurement of gas or water vapor transmission rates. These methods involve various technologies, such as calcium corrosion [1], cavity ringdown spectroscopy (CRDS) [2], tunable diode laser absorption spectroscopy (TDLAS) [3], isotope marking mass spectrometry (IMMS) [4,5], gravimetric measurement [6], and coulometric testing [7]. The optical calcium corrosion test is the most well-known method and has the advantage of detecting point defects in a barrier film, whereas the other methods measure the overall water vapor transmission rate (WVTR) of gas barrier films. The optical calcium test is also one of the most sensitive methods, allowing WVTR measurements in the range of 10^−6^ g m^−2^ day^−1^ [1]. However, it requires an extended measurement duration. On the other hand, the quadrupole mass spectrometer (QMS) technique has been demonstrated to have a high sensitivity, comparable to the calcium test method, with a greatly reduced measurement time [8,9,10].

Conventional encapsulation solutions such as metal or glass casing are known to provide excellent protection against corrosion and good biocompatibility. However, they suffer from certain limitations with regards to miniaturization and flexibility potential. In order to overcome this obstacle, advanced polymer-based barrier coatings are being developed. Poly-para-xylylene and its derivatives, commonly referred to parylenes, are polymeric films that can be deposited at ambient temperature using the Gorham process through chemical vapor deposition [11]. Due to a combination of unique material properties and process conformality, parylene encapsulation is a highly effective technology for protecting sensitive electronics, medical devices, and other products from harsh environmental factors such as moisture, heat, and chemicals. In addition, thin film encapsulation is a very promising solution for applications that necessitate flexibility, such as wearable devices and bendable displays. Flexible electronics are particularly suitable for creating comfortable wearable devices that can conform to the body’s shape, and flexible displays are increasingly popular due to their ability to be curved or rolled up, enabling new form factors. Examples of such devices include smartwatches, smartphones, e-readers, health monitoring devices, and organic light-emitting diode (OLED) panels [12,13,14,15]. In the medical domain, implantable devices, such pacemakers and catheters, requiring a high degree of biological safety including efficient protection against body fluids and maximal miniaturization, utilize parylene encapsulation to enhance their durability [16,17,18]. Due to its high biocompatibility and biostability [19,20] as well as its superior temperature stability [21,22], parylene is often cited as the material of choice for medical encapsulations [23,24,25,26].

The focus of this study was to use a newly developed permeability measurement system, based on QMS detection technique, to evaluate the permeability of four common parylene grades by quantifying the transmission rates of gases and water. The transmission rates of helium and neon were measured in order to determine their respective HTR and NTR. Furthermore, the water vapor transmission rate (WVTR) and water transmission rate (WTR) were determined in order to illustrate the difference between vapor and liquid water measurement configuration modes. The relevance of measuring WTR is particularly significant for implantable medical devices, as they are in direct contact with body fluids. The validity of the newly developed measurement system was confirmed by conducting standardized water vapor permeation measurements.

## 2. Materials and Methods

### 2.1. Gas, Water Vapor and Liquid Water Permeation System

A schematic diagram of the newly developed permeation system is illustrated in Figure 1. The system operated based on the principle of quadrupole mass spectrometry (QMS), a widely used technique for identifying and separating ionized molecules based on their mass-to-charge ratio. In this system, the QMS was used to detect the molecules passing through a barrier layer. The permeation system was divided into two main parts: the supply side, which was located ahead of the sample, and the detection side, which was behind. The supply side was responsible for delivering the gas, vapor, or liquid permeant to the barrier layer surface, while the detection side was responsible for measuring the ions diffusing through the barrier. To maintain a constant temperature, several heaters and thermocouples were placed on the tubes and chamber made of stainless steel. The inner surface of the detection chamber was electrochemically polished to reduce the amount of adsorbed gas by decreasing the specific surface area. This helps to minimize background interference and ensure accurate detection of the molecules of interest.

### 2.2. Calibration of Permeation System

Before performing measurements, the experimental setup underwent calibration using a calibrated leak composed of sintered stainless steel, previously introduced by Yoshida et al. [9,27]. This constant conductance element (CCE) leak possesses a porous structure with pore sizes less than 1 µm, which yields a constant conductance, enabling it to produce a reference molar flow for the in situ calibration of QMS detection. When gas diffuses through the CCE, molecular flow conditions are satisfied for pressures ranging from 10 to 100 mbar. Once the molecular conductance of the CCE has been calibrated using nitrogen gas, the molar flow can be calculated for different gases, as reported in Reference [28]. Equation (1) outlines how the gas flow rate (Q_gas_) (mol/s) was calculated:(1)$$Q_{\mathrm{gas}}{=C}_{N_{2}}\;\sqrt{\frac{M_{N_{2}}}{M_{\mathrm{gas}}}}\;\frac{P_{R}}{R\;\sqrt{{T\;T}_{C}}}\;,$$
where the constant ${\;C}_{N_{2}}$(m^3^/s), provided by the supplier, represents the molecular conductance of N_2_ for the CCE; $M_{N_{2}}$ and M_gas_ denote the molar mass of N_2_ (28 g/mol) and the selected gas, respectively; P_R_ (Pa) is the upstream gas pressure on the supply side, while R (J mol/K) is the gas constant; T (K) represents the temperature of the gas applied during the measurement; and T_C_ (K) is the temperature used during the calibration. This equation enables establishment of the correlation between molar flow (mol/s)—obtained by varying the upstream gas pressure, P_R_—and ion current (A) measured by the mass spectrometer, allowing for quantification of the amount of gas or vapor molecules per unit of time diffusing through the test membranes. Calibration curves for helium, neon, and water vapor are shown in Figure 2.

The gas transmission rate (GTR) (cm^3^ (STP) m^−2^ day^−1^ atm^−1^) was then calculated by:(2)$$\mathrm{GTR}=\frac{Q_{\mathrm{gas}}{\;R\;T}_{0}{\;10}^{6}\;\times \;24\;\times \;3600}{P_{\mathrm{atm}}\;A\;\Delta P\;},$$
where Q_gas_ (mol/s) is the gas flow rate calculated above; R (J mol/K) is the gas constant; T (K) and P_atm_ (Pa) are the temperature and the pressure under standard conditions, namely 273.15 K and 10^5^ Pa, respectively; A (m^2^) is the effective surface area of the membrane; and ΔP (atm) is the delta pressure between the two sides of the membrane. ΔP can be approximated by the gas supply pressure being applied on one side of the membrane, as suggested by Yoshida et al. [29].

Additionally, the water vapor transmission (WVTR) (g m^−2^ day^−1^) and the water transmission rate (WTR) (g m^−2^ day^−1^) were obtained by the following equation:(3)$$\mathrm{WVTR}\;\&\;\mathrm{WTR}=\frac{Q_{H_{2}O}{\;M}_{H_{2}O}\;24\;\times \;3600}{A\;},$$
where $Q_{H_{2}O}$ (mol/s) is the water flow rate given by the calibration curve, $M_{H_{2}O}$ is the molar mass of water (18 g/mol), and A (m^2^) the membrane area.

### 2.3. Water Vapor Permeation ISO 15106-03

The electrolytic detection sensor method using WDDG instruments (manufactured by Brugger Feinmechanik GmbH) was employed to determine the water vapor transmission rate, in accordance with the international standard ISO 15106-03. A sample with a 100 mm diameter was introduced into a test cell with two chambers: a dry chamber and a controlled-humidity chamber. The latter was equipped with a sulfuric acid solution that delivered a constant water vapor pressure. The sample’s coated side was directed towards the dry chamber, where the sensor was placed. As the water vapor permeated through the sample, it was carried by the dry nitrogen carrier gas into the electrolytic cell. This cell featured two spiral wire electrodes coated with a thin layer of phosphorous pentoxide. By applying a DC voltage, the water vapor in the carrier gas was electrolytically decomposed into hydrogen and oxygen. The mass of the permeating moisture per time interval was then determined by calculating the electrolytic current per area of the test specimen. The water vapor transmission rate was calculated by the following equation:(4)$$\mathrm{WVTR}=\frac{\;I\;}{A}\;8.067,$$

The water vapor transmission rate is denoted WVTR (g m^−2^ day^−1^). A (m^2^) represents the transmission area of the test specimen in square meters, I (A) stands for the electrolytic current in amperes, and 8.076 is the instrument constant. The measurements were conducted at a temperature of 23 °C and a relative humidity (RH) of 50%.

### 2.4. Parylene Film Deposition

The deposition of parylene film membranes was carried out through low-pressure chemical vapor deposition (LPCVD) based on the Gorham process [11] at room temperature. The parylene deposition process involved three steps: vaporizing the solid dimer in the gas phase at temperatures ranging from 80–120 °C, cleaving the dimers in monomers at temperatures between 650 and 770 °C (sublimation and pyrolysis temperatures varied for the different parylene forms), and finally, condensing and polymerizing the monomers in the chamber to form a polymeric film at a pressure of about 0.1 mbar. The parylene thickness was measured by a surface profiler (Alpha-Step^®^ 500, Tencor Instruments, Mountain View, CA, USA). The membrane thicknesses varied from 14 to 43 µm. Parylene coatings were deposited on a cleaned glass substrate in order to avoid strong adhesion of the parylene film with the substate.

Four grades of parylene were deposited and tested: parylene N, C, VT4, and AF4. Parylene N, or poly(p-xylylene), is the basic form of parylene and has a linear carbon-hydrogen molecular structure. With a low dielectric constant and a high degree of crystallinity, it is particularly suitable for high-frequency electronic applications [30,31]. Parylene C, or poly(chloro-p-xylylene), is the most widely used form of parylene. It has low permeability to gases and moisture [32] and can be deposited at a high rate [33]. Parylene VT4 and AF4, or poly(tetrafluoro-p-xylylene), are fluorinated parylene types, also known as parylene F. Parylene VT4 contains fluorine atoms in the aromatic sites, while parylene AF4 replaces the α hydrogen atoms with fluorine at the end of its aromatic ring. Fluorinated parylene types have superior thermal stability and a low dielectric constant [22,34]. Parylene AF4 is highly resistant to oxidation and UV exposure and has the highest penetrating ability among the parylene types [21,35].

### 2.5. Measurement Procedure

Self-standing parylene membranes of 40 mm diameter were placed on an O-ring sample support in supply side of the measurement system. After pumping on both sides of the sample, the system was heated up at 55 °C for at least 8 h for degassing and then cooled down at 23 °C for the measurement. In the case of gaseous permeant, a defined supply pressure up to 100 mbar was maintained in the test volume using an injection valve and a primary pump. In the case of liquid water measurement, the water was directly poured on the sample. The molecules that permeated through the film were detected by the QMS measuring a current signal (A), which was then converted first into molar flow rate (mol/s) thanks to the calibration curves and finally into transmission rates with Equations (2) and (3).

## 3. Results

### 3.1. Gas Transmission Rates

Figure 3 illustrates the normalized helium transmission rate (HTR) and neon transmission rate (NTR) results for all four types of parylene coatings. Comparing the data, it is evident that unfluorinated parylene N (PxN) and parylene C (PxC) exhibit superior hermeticity when it comes to helium and neon gases. In contrast, both parylene F types (PxVT4 & PxAF4) demonstrate HTR and NTR values approximately one order of magnitude higher. Specifically, PxC demonstrates the tightest barrier properties among the four parylene grades, showcasing an HTR and NTR of 5.1 × 10^4^ and 6.1 × 10^3^ µm cm^3^ (STP) m^−2^ day^−1^ atm^−1^, respectively.

### 3.2. Water Vapor and Water Transmission Rate

Figure 4 presents the results of measuring the water vapor transmission rate (WVTR) at a temperature of 23 °C and 50% RH using two different methods: the standardized method (ISO15106-03) and the newly developed system. The results obtained from the standardized method are highlighted in light blue, while those from the new system are displayed in the hatched, light blue columns. In addition, the water transmission rate (WTR) measured by the new system is represented by the dark blue columns. Transmission rates are normalized to the thickness of the films.

Figure 4 reports the comparison between WVTR results obtained using the WDDG instrument (ISO standard method) and those obtained through the newly developed measurement system. The results show a high level of agreement, with minimal differences observed. The maximum deviation, which is observed for parylene VT4, is only 21% between the two sets of results. Similar to the gas transmission rates discussed earlier, parylene C (PxC) demonstrates excellent barrier performance when it comes to both vapor and liquid water. It exhibits a WVTR of 15.2 µm g m^−2^ day^−1^ and a WTR of 72.5 µm g m^−2^ day^−1^. These values highlight the remarkable ability of parylene C to hinder the ingress of vapor and liquid water. In contrast, parylene N (PxN) appears to be the most permeable parylene layer to water ingress, indicating relatively lower barrier properties compared to other parylene grades. This finding contradicts the earlier results on gas transmission rates, where parylene N exhibited superior barrier performance compared to parylene VT4 and parylene AF4. Interestingly, the fluorinated parylenes (PxVT4 and PxAF4) did not exhibit the worst barrier properties in terms of WVTR and WTR. This indicates that the behavior of these materials varies between gas diffusion and water permeation, emphasizing the necessity for complete barrier properties evaluations that include various permeants. Furthermore, comparing the WVTR and WTR values for each parylene type allows us to extract the acceleration factor for the transmission rate from vapor to liquid water. The accelerated factor varies between 4 for parylene N and 4.8 for parylene C. This information provides insights into the difference between water vapor and liquid water transmission through the parylene coatings.

## 4. Discussion

As demonstrated in former studies, the permeation measurement system using the QMS detection method allows for a precise quantification of gas and water vapor diffusion [8,9,10]. Based on a similar detection method, a newly developed permeation system was first used to measure helium (HTR), neon (NTR), and water vapor (WVTR) transmission rates of four parylene grades. HTR and NTR demonstrated similar trends across all types of parylene, despite helium and neon both being noble gases. However, their transmission rates differ due to variations in their atomic properties. First, neon possesses a larger atomic size compared to helium, measuring 0.26 nm and 0.22 nm, respectively [36]. This disparity in size could account for the lower NTR in comparison to HTR. Additionally, the larger size of neon atoms enables them to interact more readily with residual gas molecules, leading to higher collision rates. Consequently, those interactions can reduce the number of particles coming into contact with the membrane. The second aspect to consider is the difference in atomic mass; helium has an atomic mass of 4.003 amu, while neon’s is 20.179 amu. This difference results in higher average velocities for lighter gas molecules at a given temperature, leading to a higher “bombardment rate” of helium atoms with the membrane and subsequently increasing the transmission rates. Although there is limited literature available on neon diffusion in polymers, our findings find support in a study conducted by Nörenberg et al. [37]. In their experiment, they quantified the diffusion of noble gases through PET membranes with a thickness of 12 μm using a similar detection method. The results revealed that the transmission rate of neon was 10 times lower compared to helium, specifically 2.5 × 10^3^ cm^3^ (STP) m^−2^ day^−1^ atm^−1^ for neon and 2.2 × 10^4^ cm^3^ (STP) m^−2^ day^−1^ atm^−1^ for helium.

The comparison between the obtained WVTR values and those measured by a standardized method enabled us to validate the new instrument. Moreover, the results showed that gas and water vapor transmission rates are not following the same trends. Indeed, fluorinated parylene types (PxVT4 and PxAF4) demonstrated a lower hermeticity to helium and neon than unfluorinated parylene types (PxN and PxC). On the other hand, PxVT4 and PxAF4 exhibited better WVTR and WTR values than PxN. As introduced by Cussler et al. [38], the condensable nature of the water makes a fundamental difference to gases. Generally, it allows much higher water concentrations in the layer compared to gases, by the mechanism of capillary condensation, which raises the permeability. Furthermore, the high polarity of the water molecule could explain why fluorinated parylenes have a better WVTR than PxN. A high polarity means a high tendency for wetting polar surfaces, known to be hydrophilic, which again increases the tendency for condensation. In relation to the obtained results, fluorinated PxAF4 presents a more hydrophobic surface than PxN with measured contact angles of 105° [39] and 73° [40], respectively.

The QMS detection method offers several advantages, including short measurement duration and high sensitivity compared to other techniques. However, it also has some limitations that need to be addressed. One such limitation is the complexity of calibrating the mass spectrometer for accurate measurements. Precise control of instrument parameters is required, as deviations in calibration can affect measurement accuracy. To ensure consistent and precise results, regular calibration is highly recommended. Additionally, the sensitivity of the mass spectrometer plays a crucial role in measurement resolution. To improve the minimum detectable transmission rate, a Faraday cup with a microchannel plate electron multiplier can be utilized. This combination increases the detected current signal, thereby enhancing sensitivity. Another limitation of the QMS detection system is the presence of residual gas molecules in the vacuum system. These residual gases can generate background signals that may overlap with the signals of interest, making it challenging to accurately discern the desired information. One common residual gas is water, which contributes to the background signal. To mitigate this effect and reduce background interference, an extensive pumping phase at high temperature before the measurement is necessary to evaporate residual water in the detection chamber. An alternative solution to improve the detection limit is to replace water with deuterium oxide as the permeant. By doing so, the deuterium oxide’s mass-to-charge ratio (*m*/*z*) becomes 20 instead of 18 for water, allowing for better differentiation between the desired detection signal and the background signal.

The ability to measure the WTR of the newly developed system is a crucial factor in characterizing encapsulation materials used in the medical field. While an accurate and reliable measurement of WVTR is important in determining the shelf life, stability, and quality of products, WTR values give a more representative barrier property in the case of implantation where the devices are directly in contact or immersed in body fluids. As parylene is widely used for medical device encapsulation, the characterization of the barrier properties with liquid water is particularly relevant. Furthermore, the ability to measure GTR, WVTR, and WTR on the same system is a significant advancement in accurately characterizing the barrier properties of encapsulation materials. While the complexity of correlating gas and water vapor transmission rates underscores the importance of measuring WVTR and GTR provides a complete characterization of the material’s barrier properties. In summary, accurate measurement of WTR values is essential in the medical field, particularly for implantable devices, but measuring all three transmission rates on the same system allows for a more comprehensive assessment of encapsulation materials. This is particularly important for materials like parylene film that are commonly used for medical device encapsulation, where the ability to resist the diffusion of liquids, gases, and vapors is crucial for ensuring the long-term stability and performance of the device.

## 5. Conclusions

To conclude, the utilization of a recently developed permeation measurement system employing the QMS detection method has proven to be effective in determining gas and water vapor transmission rates (WVTR). The obtained results were compared against a standardized method in order to validate the WVTR measurements of thin parylene films. While WVTR measurement holds significance in assessing barrier performance across various fields, the evaluation of water transmission rate (WTR) is of greater importance when evaluating barrier films for medical device encapsulation. This study has provided valuable insights into the application of permeation measurement systems, especially in medical contexts. The incorporation of gas, vapor, and liquid water transmission rate measurements within a single system has been demonstrated to be advantageous. As establishing correlations between gas and water vapor transmission rates can be challenging, the added value of measuring gas, vapor, and liquid water transmission rates using a single system is well-established and widely acknowledged. Finally, the findings of this study contributed to the understanding of permeation measurement systems to a certain level and highlighted the relevance of such measurements especially in medical applications.

## Figures and Tables

**Figure 1 polymers-15-02557-f001:**
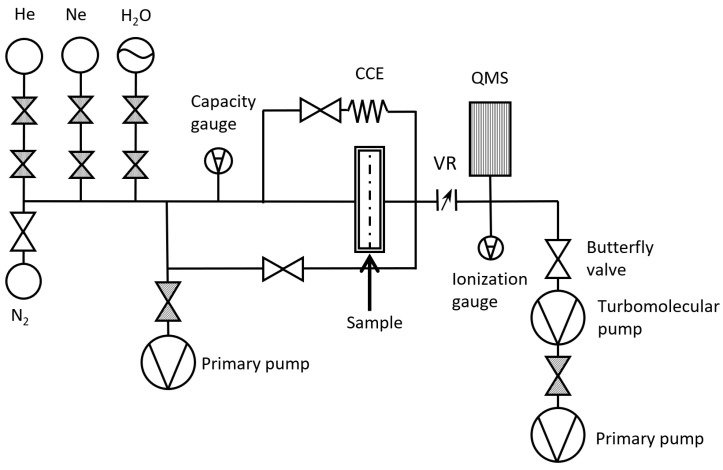
Schematic diagram of the newly developed permeation measurement system.

**Figure 2 polymers-15-02557-f002:**
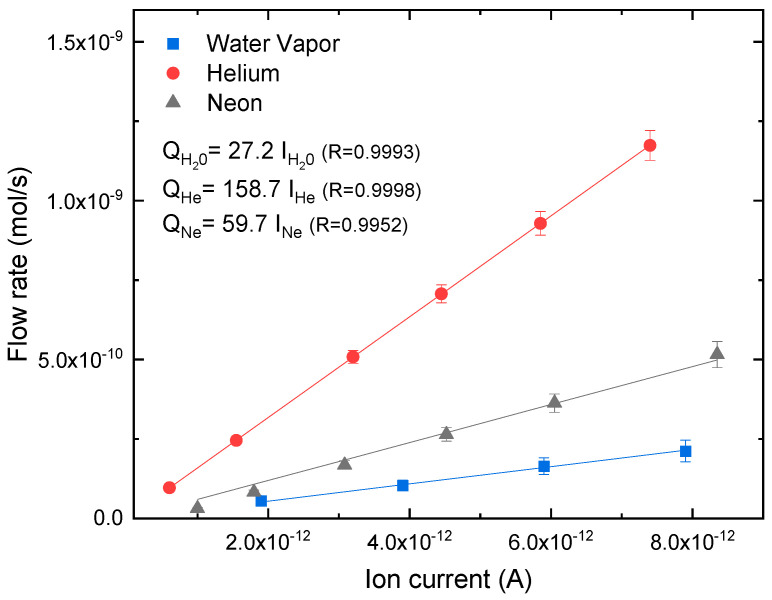
Calibration curves for helium, neon, and water vapor which allow for the relationship between the ion current (A) from the QMS and the flow rate (mol/s) to be obtained.

**Figure 3 polymers-15-02557-f003:**
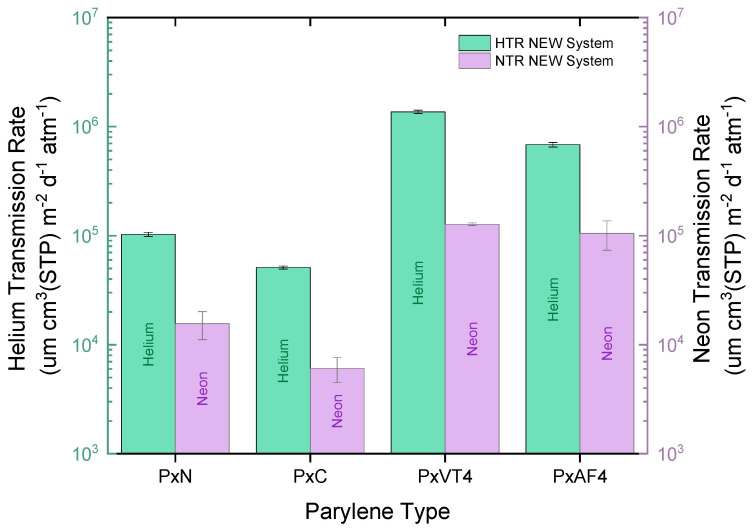
Normalized HTR and NTR (µm cm^3^ (STP) m^−2^ day^−1^ atm^−1^) of parylene N, C, VT4, and AF4, respectively PxN, PxC, PxVT4, and PxVT4 in the figure. Highest hermeticities regarding helium and neon are exhibited by PxC with an HTR and NTR of 5.1 × 10^4^ and 6.1 × 10^3^ µm cm^3^ (STP) m^−2^ day^−1^ atm^−1^, respectively.

**Figure 4 polymers-15-02557-f004:**
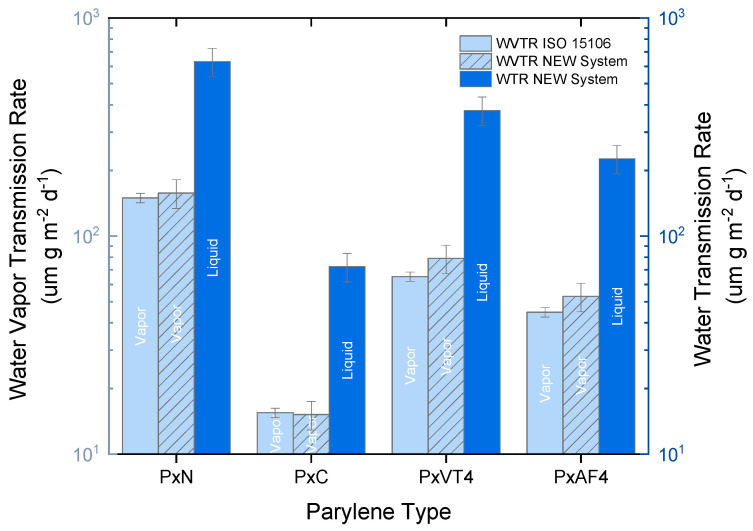
Normalized WVTR and WTR (µm g m^−2^ day^−1^) values for all four parylene types. Similar to gas transmission rate results, parylene (PxC) exhibits the best barrier performance regarding water diffusion with a WTR of 72.5 µm g m^−2^ day^−1^.

## Data Availability

The data presented in this study are available on request from the corresponding author.

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
