# Peer review of "Development of a Water Transmission Rate (WTR) Measurement System for Implantable Barrier Coatings"

_polymers, 2023, doi:10.3390/polym15112557_

Round 1
Reviewer 1 Report
The authors developed a new permeation measurement system for detecting gas and water vapor transmission rates of implantable barrier coatings successfully. I have the following questions about the content of the manuscript to discuss with the authors.
(1) According to materials and methods, four types of parylene films were deposited using LPCVD. Have the authors considered detailed characterization of the deposited-films, such as roughness, film thickness, morphology, etc. I believe this will make the manuscript more complete.
(2) In Figure 4, why not use the standardized method (ISO15106-03) to measure water transmission rate of deposited films to prove the effectiveness of newly developed systems?
(3) The very latest references (e.g., Acta Materialia 232 (2022) 117934; Journal of Materials Science & Technology 134 (2023) 33–41) are lacking, which may be helpful for introduction and/or discussion?
(4) Overall, little content of the results section was presented and it is proposed to further enrich it. Can a more other methods be used to measure the transmission rate of thin films or more analysis of gas/water penetration abilities for different parylene films?
Moderate editing of English language
Author Response
Dear reviewer,
Thank you very much for your comments. Please find enclosed, the Word document containing our replies.
Best regards,
Sébastien Buchwalder

Reviewer 2 Report
The paper is well-written and effectively describes the experiments conducted. The authors have created a methodology for determining the GTR and WTR of coating materials, such as parylene. I do have two minor suggestions for the authors. Firstly, it would be helpful if they could state any limitations of their methods, such as whether a free-standing film is required for coating. Secondly, it would be beneficial if they could explain why they expect neon to have a lower TR than helium and why water is expected to have the lowest TR in their calibration curve.
Good. Would need grammar check
Author Response

(The authors gave the same response as above.)
